# HN1 Is Enriched in the S-Phase, Phosphorylated in Mitosis, and Contributes to Cyclin B1 Degradation in Prostate Cancer Cells

**DOI:** 10.3390/biology12020189

**Published:** 2023-01-26

**Authors:** Aadil Javed, Gülseren Özduman, Lokman Varışlı, Bilge Esin Öztürk, Kemal Sami Korkmaz

**Affiliations:** Cancer Biology Laboratory, Department of Bioengineering, Faculty of Engineering, Ege University, Bornova, 35040 Izmir, Turkey

**Keywords:** cell cycle, mitosis, HN1, kinases, prostate cancer, stable cell line

## Abstract

**Simple Summary:**

Hematological and Neurological Expressed 1 (*HN1*) has previously been explored in Prostate cancer, where it was highly expressed as compared to normal Prostate cells. The HN1 co-expression network in Prostate cancer displayed two distinct nodes of G1/S transition-related genes and mitotic protein encoding genes. Interestingly, HN1 expression was found as inversely correlated with Cyclin B1. The mechanistic data for the involvement of HN1 in the cell cycle dynamics and pathways are not available. Here, we employed cell cycle synchronizations coupled with kinase inhibitors and overexpression experiments. HN1 levels fluctuated in the cell cycle with enriched levels in the S-phase, with a distinct phospho-HN1 form appearing in mitosis, which disappeared upon treatment with inhibitors of GSK3β and Cdk1 kinases. Mechanistically, HN1 interacted with Cdh1 (a co-factor of APC/C) for its stabilization and degradation of Cyclin B1, via increased ubiquitination. HN1 levels are tightly regulated throughout the cell cycle, as before Nocodazole blocks HN1 overexpression, it led to increased accumulation of cells in the S-phase and after Nocodazole arrest, it led to early mitotic exit. Therefore, HN1 is a novel cell cycle-regulated protein with potentially dual roles, involved in the modulation of cell cycle progression at the S-phase and mitotic exit.

**Abstract:**

HN1 has previously been shown as overexpressed in various cancers. In Prostate cancer, it regulates AR signaling and centrosome-related functions. Previously, in two different studies, HN1 expression has been observed as inversely correlated with Cyclin B1. However, HN1 interacting partners and the role of HN1 interactions in cell cycle pathways have not been completely elucidated. Therefore, we used Prostate cancer cell lines again and utilized both transient and stable inducible overexpression systems to delineate the role of HN1 in the cell cycle. HN1 characterization was performed using treatments of kinase inhibitors, western blotting, flow cytometry, immunofluorescence, cellular fractionation, and immunoprecipitation approaches. Our findings suggest that HN1 overexpression before mitosis (post-G2), using both transient and stable expression systems, leads to S-phase accumulation and causes early mitotic exit after post-G2 overexpression. Mechanistically, HN1 interacted with Cyclin B1 and increased its degradation via ubiquitination through stabilized Cdh1, which is a co-factor of the APC/C complex. Stably HN1-expressing cells exhibited a reduced Cdt1 loading onto chromatin, demonstrating an exit from a G1 to S phenotype. We found HN1 and Cdh1 interaction as a new regulator of the Cyclin B1/CDK1 axis in mitotic regulation which can be explored further to dissect the roles of HN1 in the cell cycle.

## 1. Introduction

Prostate cancer is a major health concern worldwide, particularly among men [1]. There are several proteins, which are overexpressed in Prostate cancer, associated with cell cycle-regulating pathways [2,3]. Cell cycle regulatory proteins are key targets for therapeutic interventions in various cancers [4]. Cell cycle processes have been defined in detail; however, the complexity of the pathways and numerous unknown factors underlie the ongoing studies to further our understanding of the life cycle of cells, especially in the context of cancer [5].

Hematological and neurological expressed 1 (*HN1*) is a gene located on chromosome 17 in humans with the conserved protein encoded across vertebrates, and is also referred to as Jupiter microtubule homolog 1 (JPT1) due to sequence similarity with the Jupiter protein in Drosophila Melanogaster [6,7]. *HN1* has been identified as a novel gene that is associated with castration-resistant prostate cancer (CRPC) in a study that analyzed the transcriptome of human LNCaP cancer cells while progressing to CRPC. Higher expression of *HN1* was detected along with 14 other genes that were associated with metastasis of prostate cancer [8]. Expression of HN1 has been reported in prostate cancer cell lines including LNCaP, DU-145, and PC3. It was revealed that HN1 expression was responsive to the epidermal growth factor (EGF) in LNCaP and PC3 cells, while its depletion in PC3 cells resulted in increased pGSK3β^S9^ and Cyclin B1 along with the accumulation of β-catenin and prolongation of the G1 phase of the cell cycle [9,10,11]. Since HN1, AR, and EGFR have been reported as highly expressed genes in prostate adenocarcinoma and stimulation of AR without androgens gives a survival advantage to prostate cancer cells post-androgen ablation therapy, HN1 could be placed at a crossroads in intervention therapies that involve targeting AR in cancers’ refractory to hormones [12]. Recently, HN1 was shown to exhibit a novel interaction with γ-tubulin for the regulation of centrosome-related processes in advanced Prostate cancer cells [13]. Therefore, it is necessary to investigate HN1 and its role in cell cycle regulation.

Anaphase Promoting Complex/Cyclosome (APC/C) is a complex E3 ubiquitin ligase complex, which regulates major cell cycle processes. Its activity depends on two major co-factors, namely Cdc20 and Cdh1 [14]. APC/C upon activation from Cdh1 leads to Cyclin B1 proteasome-mediated degradation by increasing ubiquitination [15,16]. Cyclin Dependent Kinase 1 (CDK1) is activated due to a complex formation with Cyclin B1 and carries out the G2/M transition and mitotic processes via activation and inhibition of its substrates [17,18]. On previous occasions, HN1 depletion led to the accumulation of Cyclin B1 [9] and its overexpression decreased Cyclin B1 [9,19]. The effect of this inverse correlation of HN1 and Cyclin B1 on cell cycle dynamics remains unknown.

In this study, we examined the native expression of HN1 in different phases of the cell cycle by synchronizing cells in different phases. Furthermore, we showed that HN1 is a cell cycle regulatory protein as its overexpression in synchronized cells changed the cell cycle dynamics and increased the degradation of Cyclin B1, which is one of the major targets of APC/C mediated proteasomal degradation [20,21]. Since Cyclin B1 has important functions in mitotic progression [22], we explored the varying expression levels of HN1, and whether it has a mechanistic contribution to the Cdh1/Cyclin B1 pathway. The data presented here demonstrated that HN1 is a cell cycle regulatory protein with tight control over its expression throughout the cell cycle and it interacts with cell cycle mediators, which has not been studied in detail previously.

## 2. Materials and Methods

### 2.1. Mammalian Cell Culture

PC3 and DU-145 cells (from American Type Culture Collection-ATCC) were grown in DMEM-F12 (Invitrogen, Waltham, MA, USA) media supplemented with inactivated 5% fetal bovine serum (FBS, Sigma, Welwyn Garden City, UK) and additives L-glutamine (2 mM, Invitrogen, Waltham MA, USA), penicillin (100 U/mL), and streptomycin (100 µg/mL, Invitrogen, Waltham MA, USA). The aforementioned cell lines were cultured at 37 °C in a humidified atmosphere with 5% CO_2_.

### 2.2. Chemicals and Antibodies

LY294002 (PI3K kinase inhibitor), SB216763 (GSK3-β inhibitor), RO-3306 (Cyclin-dependent kinase 1 inhibitor), Flavopiridol (pan-CDK inhibitor), TAME (N2-[(4-Methylphenyl)sulfonyl]-L-arginine methyl ester) (APC/CCdc20 inhibitor), AG556 (EGFR inhibitor), CAS 587871-26-9 (ATM inhibitor), Wortmannin (PI3K/Akt pathway inhibitor), PD98059 (MAPK inhibitor), and TBCA (Casein Kinase II inhibitor) were purchased from Sigma Aldrich (Berlin, Germany). Antibodies against HN1 were either produced as previously described [13] or purchased from Invitrogen (Waltham MA, USA). β-actin antibodies were purchased from Sigma (Berlin, Germany). Antibodies against Cyclin B1, Cdk1, p-H3^(S10)^, Cdt1, control IgG (mouse and rabbit), Ubiquitin, Cdc20, Emi1, BrdU, GAPDH, Lamin B1, Acetylated H4, and Cdh1 were either purchased from Sigma Aldrich, Invitrogen, or Santa Cruz Biotechnology (Heidelberg, Germany). Anti-mouse and anti-rabbit AlexaFluor, 488- and 594-conjugated antibodies, were purchased from Invitrogen (Carlsbad, USA). The antibodies were used at concentrations of 0.2 to 1 µg/mL.

### 2.3. Cell Cycle Synchronizations

Cells were treated with cell cycle inhibitors to induce arrest at S or G2/M (prometaphase) using 2 mM Thymidine (Sigma Aldrich, Berlin, Germany) (2 blocks) and/or 165 nM Nocodazole (Sigma Aldrich, Berlin, Germany) (at 16 to 24 h), respectively [23,24]. To collect cells in the S phase, cells were released into a fresh medium after a double Thymidine block for 6 h. To determine the temporal response of mitosis to G1 entry (mitotic exit), cells were released from the Nocodazole block into G1 for more than 2 h. Serum starvation was carried out by culturing the cells in serum-free media for 48 h to enrich cells in the G0/G1 phase of the cell cycle with other culture conditions maintained at normal levels [25]. The mitotic shake-off approach was utilized to enrich and harvest cells undergoing mitosis by treating cells with Nocodazole for 16–18 h, released in fresh media for 1 h, and collected by tapping the culture dishes [26].

### 2.4. Plasmid DNA and siRNA Transfections

HN1 ORF was inserted into the pcDNA4-HisMax backbone as described previously [13]. Plasmid DNA transfections were performed with FUGENE-HD (Cat. No. E2311) transfection reagent (Madison WI, USA). Cells seeded on 6-cm dishes were treated with a mixture of 1 μg DNA and 3 μL transfection reagent prepared in 100 μL full negative DMEM media (the mixture was incubated at room temperature for 15 min and added dropwise to the cells). Cells were harvested by either cell scraping or trypsinization. If the downstream application was flow cytometry, cells were collected by trypsinization. HN1 overexpression using HM-HN1 plasmid transfection (as a comparison with vector (HM-Vec) transfection) was performed in previous studies as well. siRNA specifically targeting HN1 was used for HN1 depletion as described previously. The off-target effects along with specific targeting efficacy and transfection efficiency for siRNA and scrambled control RNA had been validated [13]. Briefly, cells were plated on 6-cm dishes and after 24 h, media was replaced with antibiotic-free media 4 h before transfection. A mixture of either scrambled RNA (sc-37007) and/or siHN1 (sc-93940) (Dallas TX, USA) was prepared with FUGENE-HD reagent (1:1) and incubated at room temperature for 15 min. The complexes were added to cells dropwise without disturbing the monolayer of the cells. The minimum duration of transfection was kept at 48 h and cells were harvested according to the method suitable for downstream application.

### 2.5. Stable Cell Line Generation

SmaI and EcoRV (New England Biolabs (NEB), Hitchin, UK) enzymes were utilized to digest out the HN1 ORF from pcDNA4-HisMax-HN1 and inserted into pCW57.1 (a gift from Dr. Şerif Şentürk at IBG, Izmir, Turkey) which was prepared by deleting the ccdB fragment by using SalI (NEB) and NheI (NEB) enzymes. Blunt end ligation was performed using T4 DNA ligase (NEB), followed by the transformation into chemically competent DH5α cells. The pCW57-HM-HN1 plasmid was obtained after isolation using the Midi-prep kit (Qiagen, Germany). pCW57-HM-HN1 as a transfer vector along with the viral packaging plasmids obtained from Addgene such as pMDLg/pRRE, pRSV-Rev, and pMD2G were transfected into HEK293T cells grown in DMEM/F12 media (supplemented with 5% FBS) and normal cell culture conditions using PEI (1:3 DNA to reagent ratio optimized) as a transfection reagent (Transporter 5, Polysciences, Germany). Chloroquine (Sigma Aldrich, Europe) at the final concentration of 25 μM was added to the culture media to block endosome formation and increase the transfection efficiency. The viral supernatant was harvested at 72 h of post-transfection, pelleted by centrifugation at 16,000× *g* overnight, and re-suspended in 100 μL PBS solution. Polybrene (Sigma Aldrich, Europe) at a final concentration of 10 μg/mL was used to transduce target cells with pCW57-HM-HN1 lentiviral particles. Twenty-four to 48 h after transduction, the cells were allowed to grow in normal conditions for 24 h before puromycin (Sigma Aldrich, Europe) selection (2μg/mL) for six days (determined after performing a kill-curve with titrating Puromycin concentrations from 1 to 10μg/mL). Optimized 1 μg/mL Doxycycline (Sigma Aldrich, Europe) was used for inducing HN1 expression in HN1-inducible stable cells.

HN1 ORF was fused with a Venus fluorescence expressing tag sequence to determine the HN1 localization within Doxycycline inducible cells. Briefly, HN1 was amplified using a forward primer with an EcoR1 site and reverse primer (without Stop codon) with an Not1 site, and inserted into a pcDNA4-TO-Venus-Puro plasmid obtained from Addgene (44118). Moreover, the HN1-Venus fragment was then digested and inserted into the pCW57 backbone to generate a Tetracycline inducible HN1-Venus construct, from which stabilized PC3 cells were generated after lentiviral transduction and antibiotic selection as described above.

### 2.6. Protein Isolation, Immunoblotting, and Immunoprecipitations

The ice-cold RIPA buffer was used for cell lysis as described previously [13]. The recipe of the buffer contains 1% Nonidet P-40, 50 mM Tris-HCl (pH 7.4), 0.25% Na-deoxycholate, and 150 mM NaCl with 1 mM NaF, 1 mM EDTA, 1 mM Na3VO4, and complete protease along with phosphatase inhibitors and cocktails (Roche, Munich, Germany). After washing the cell pellet with PBS, cells were incubated with a RIPA lysis buffer and placed on ice for 45 min before sonication at 20 s with 25% power for 50 cycles. The total protein lysate was collected from the supernatant after centrifugation at 12,000× *g* for 10 min at 4 °C. Protein quantitation was performed based on the BCA assay kit (Sigma Aldrich, Berlin, Germany). For the Lambda protein phosphatase assay, λ Protein phosphatase (Lambda PP: NEB, Cat no: P0753S) (Ipswich MA, USA) was applied to the lysates with the manufacturer’s recommendations. The λ Protein phosphatase treated lysates were not subjected to the phosphatase inhibitor treatment used for the protein isolation buffer.

SDS-polyacrylamide (10–15%) gels were used for separating proteins using electrophoresis and a wet transfer blotter to immobilize PVDF membranes (Amersham, Buckinghamshire, UK or Millipore, Livingston, Scotland, UK). Then, membranes were blocked in TBS-T (Tris-buffered saline containing 0.1% Tween-20) containing a 5% milk (*w*/*v*) solution. The specific antibody staining for immunoblotting was performed by incubating antibodies prepared in appropriate dilutions (0.5 percent milk in TBS-T) at room temperature for one hour, followed by washings (4 times) in a TBS-T buffer and secondary antibody incubation. The membranes were washed again after secondary antibody incubation. The ECL Plus reagent (Amersham, UK) was prepared according to the manufacturer’s recommendations and used for chemiluminescence on the PVDF membrane for 5-min incubation. The Kodak X-ray films were exposed to membranes and then developed in the darkroom using standard developer and fixer solutions.

Protein lysates were obtained for immunoprecipitation as mentioned for immunoblotting. The amount of protein lysate used for each immunoprecipitation reaction was 0.5 mg with one-hour preclearance performed in the IP-matrix (40µL) obtained from Santa Cruz Biotechnology Inc. or Dynabeads protein G (10003D, Thermo Fischer Scientific, Waltham, MA, USA) and used according to the manufacturer’s recommendations. Briefly, lysates divided into two were used for the reaction with either a specific antibody against a protein of interest or a non-specific immunoglobulin G for 4 h at 4 °C. After the reaction, the IP matrix (40 µL) or magnetic beads (20 µL) were used to form complexes with the antibody in the reaction of the antibody and the lysate overnight at 4 °C before washing with a RIPA-modified buffer. The complexes were washed thoroughly in a RIPA-modified buffer four times before denaturation of the samples in a Laemli buffer (25 µL) for 5 min at 95 °C. The solution was then run on SDS gel for protein separation and immunoblotted for target antibodies as specified previously [13].

### 2.7. Sub-Cellular Fractionations

Sub-cellular fractionations required cells to be washed in PBS and centrifuged at 300× *g* for collecting cell pellets, which were re-suspended in Buffer A (10 mM KCl, 50 mM HEPES pH: 7.4, 1 mM EDTA, 1 mM EGTA) and shaken on a rotator for 30 min at 4 °C. After shaking, lysates were centrifuged at 4000× *g* for 5 min and the supernatant was collected at a cytosolic or cytoplasmic fraction, whereas the pellet was further processed for nuclear fraction isolation. The pellet was washed four times in Buffer A to remove any cytoplasmic proteins with subsequent centrifugations at 4000× *g* for 5 min at 4 °C. Nuclear pellets were re-suspended in Buffer B (400 mM KCl, 1M HEPES pH: 7.4, 1 mM EDTA, 1 mM EGTA, 0.5% Triton-X-100) and placed on a rotator for constant shaking for 30 min at 4 °C. Then, centrifugation was performed at 14,000× *g* for 30 min at 4 °C to collect nuclear fraction from the supernatant. For chromatin fraction, the RIPA modified buffer was added to the pellet and the fraction was re-suspended and sonicated for 20 s with 25% power for 50 cycles. Protein quantitation was performed based on the BCA assay kit (Sigma, UK).

### 2.8. Immunofluorescence (IF) Labeling and Microscopy

For immunofluorescence labeling and subsequent microscopy, after concluding the specific treatments, cells grown on coverslips were fixed in either cold methanol (100%) at −20 °C for 30 min or paraformaldehyde (4% in PBS) at room temperature for 30 min, were washed with PBS, and were permeabilized in 0.2% Triton X-100 in PBS for 5 min at the shaker, before another washing. The 1% Bovine Serum Albumin (BSA) prepared in PBS was used for blocking the coverslip for 5 min of incubation on a shaker at room temperature. Primary antibody dilutions were prepared in 1% BSA in PBS and added to the cells followed by incubation for one hour in a humidified chamber at 37 °C. Cells were washed with PBS four times before secondary antibody incubation for 20 min in the same conditions as primary antibody incubations. The secondary antibodies were either AlexaFluor 594 (anti-mouse)- or AlexaFluor 488 (anti-rabbit)-conjugated antibodies (Invitrogen, Carlsbad, CA). After incubation, cells were washed with PBS four times and treated with 70% ethanol for one minute and 100% ethanol for one minute. The cover glasses were then air-dried and mounted on glass slides with 0.5 to 1 µg/mL DAPI in 30% glycerol in PBS and analyzed immediately under a DM4000 LED B fluorescence microscope (Leica) as described previously [13]. Leica imaging software was used with a 5.5 Mpix digital camera for capturing images.

### 2.9. Flow Cytometry Analysis

To determine the cell cycle distribution upon treatment with various inhibitors and transfections, propidium iodide (PI) staining was performed on the cells harvested by trypsinization and fixed in 70% ethanol in PBS, and stored at −20 °C for at least 24 h. Briefly, fixed cells were washed with PBS and incubated with 0.2% Triton X-100 in PBS for 5 min at room temperature on a shaker. The cells were pelleted by resuspending in 20 µg/mL RNAse A solution in PBS and incubated at 37 °C for 30 min. Cells were pelleted again and resuspended in 1 µg/mL PI in PBS. Afterward, the cells were analyzed for DNA content on the C6 BD Accuri flow cytometer (Becton Dickinson, Franklin Lakes, NJ, USA) and analyzed for cell cycle distribution on ModFit LTTM or Flowjo v10 software.

### 2.10. Statistical Analyses

The data values are presented as the means ± standard error of the mean (SEM). The statistical analysis was conducted on either Microsoft Excel or GraphPad Prism 5 software packages. Differences in mean values between groups were analyzed using a two-tailed Student’s *t*-test or ANOVA, where *p* < 0.05 was considered statistically significant.

## 3. Results

### 3.1. HN1 Protein Levels Fluctuate during the Cell Cycle with a Specific Phosphorylated Form in Mitosis

HN1 levels in the cell cycle synchronized populations of PC3 cells in G1, S, prometaphase (after G2 using Nocodazole arrest), and during mitosis were observed using immunoblotting (Figure 1A). In G1 populations starved from serum for 48 h, HN1 levels were lowest as compared to other phases. Serum starvation is a method used to starve cells from mitogens and leads to cells synchronized in G0/G1 [25]. In cells blocked at the G1/S boundary using a double Thymidine block, HN1 expression increased as compared to G1 synchronized cells. In cells released into the later S-phase for 6 h using fresh media after the double Thymidine block, the HN1 expression level peaked. The 6-h release sample led to the accumulation of cells in the late S-phase; also, cells approached G2 as well. In post-G2 synchronized populations observed after Nocodazole blockage for 18 h, HN1 levels decreased but remained higher than in the G1 population (serum-starved). Interestingly, cells collected using the mitotic shake-off method exhibited a distinct higher extra band than the native HN1 band, which was not observed in lysates from other enriched cell cycle phases (Figure 1A). The flow cytometry analysis of PI-stained cells also validated the cell cycle synchronizations (Figure 1A). The HN1 protein levels normalized on beta-actin levels are shown in Figure 1B.

In PC3 cells, mitotic cells exhibited a slightly larger band than the native HN1 protein size. This was only observed in mitotic (shake-off) cells and was absent in other cell cycle-enriched phases. The overall approach of the experiment is to observe the progression of the cell cycle from G1 to mitosis. This experiment was repeated in other cell lines and provided a similar conclusion that HN1 levels fluctuate during the cell cycle with enriched levels in the S-phase, as observed in LNCaP and PC-3 cells where 2-h release from the Thymidine double block was utilized to observe early S-phase enrichment (Appendix A).

To determine whether the larger band (around 25 kDa) is a phosphorylated form of HN1, Lambda (λ) phosphatase treatment was performed in mitotic cell lysates, and the phospho-HN1 form was observed to be depleted upon phosphatase treatment, hence suggesting that HN1 is partially phosphorylated in mitosis (Figure 2A). Native HN1 also keeps up the original expression level at mitosis, suggesting that the HN1 expression and its phosphorylations are tightly regulated at mitotic transition and further in the cell cycle. Furthermore, different kinase inhibitors were applied to mitotic populations collected, as shown in Figure 1A and Appendix A, to determine the upstream kinases of HN1. The inhibitor treatment duration was 3 h in the Nocodazole block and 1-h release (total 4 h of inhibitor treatment) to explicitly inhibit the kinases in mitosis. Specific kinase inhibitors for GSK3β, EGFR, MAPK, AKT, and ATM were used and the GSK3β inhibitor treatment led to a significant depletion of the upper band of HN1, suggesting that HN1 is phosphorylated in mitosis by GSK3β (Figure 2B). Moreover, the effect of the inhibitors of PI3K, Cdk1, Pan-Cdks, along with APC/C inhibitor (TAME), and Casein kinase II with the GSK3β inhibitor as a control was also determined using the same procedure as described in Figure 1C. It was observed that the Cdk1 inhibitor (RO-3306, 10 μM) completely depleted the native and phosphorylated forms of HN1 (Figure 2C). The Pan-Cdk inhibitor (Flavopiridol, 500 nM) also led to a significant decrease in the HN1 phospho-form. These experiments were repeated and it was observed that RO-3306, even at lower concentrations, also led to a clear reduction in the expression of the HN1 phospho-form (Appendix A). CDK1i treatment of PC3 cells led to a significant decrease in HN1 expression during mitosis. Therefore, CDK1 kinase-related phosphorylation of HN1 could be functional and has to be explored further.

### 3.2. HN1 Overexpression Perturbs Cell Cycle Dynamics in Synchronized Cells

After determining that HN1 levels fluctuate in the cell cycle and remain higher in the S phase and decrease afterward, we aimed to focus on Nocodazole arrest as a cut-off point to determine the impact of HN1 overexpression. Therefore, we performed plasmid DNA transfections to overexpress HN1 before and after Nocodazole arrest (Figure 3 and Figure 4, respectively). The overexpression of HN1 with the pcDNA4-HM-HN1 construct counteracted Cyclin B1 expression level regardless of the time since Nocodazole release. The phosphorylation of histone H3 at serine 10 (p-H3^(S10)^), an indicator of mitosis, was also downregulated (Figure 3A). Intriguingly, flow cytometry analysis revealed that the cells with HN1 overexpression were arrested at the S phase when the PC3 cells were transfected before the Nocodazole block (Figure 3B). The raw data for the PI analysis is shown (Appendix A). Here, p-H3^(S10)^ levels indicated that HN1 overexpressed cells could not reach G2-M (4N population) on time as compared to control cells and were delayed at the S-phase as observed in the flow cytometry analysis.

Since HN1 is a relatively short-lived protein in PC3 cells as compared to other Prostate cancer cells (Appendix A), the impact of HN1 overexpression using the ALL-IN-ONE Tetracycline inducible expression system on cell cycle dynamics in PC3 cells was observed. Here, the ectopic HN1 expression was induced using Doxycycline (1 μg/mL) before the Nocodazole block, and induction was validated using immunoblotting (Figure 3C). Flow cytometric analysis using PI staining demonstrated that HN1 overexpressed cells accumulated more in the S phase as compared to control cells (Figure 3D). In Nocodazole arrested cells, where HN1 was induced before the block, cells remained enriched in the S phase, and cells released into mitosis after the Nocodazole block for 4 h; HN1 overexpressed cells reached G2-M (indicated by 4N population) slower than control cells (Figure 3D).

The effect of HN1 overexpression on the exit from mitosis was measured. PC3 cells were transfected with either HM-HN1 or HM-Vector during the Nocodazole block (after cells reached prometaphase (post-G2) at the 16-h mark) (Figure 4A). Since the HN1 level decreases after the S and G2 phases (Figure 1A), we aimed to overexpress HN1 at the time point when it solely lowered in the cell cycle. Following transfection, the cells were released from Nocodazole at different times (0–8 h) by adding the fresh medium, and samples were prepared and analyzed by western blotting. Cyclin B1 levels again showed a prominent decreasing trend with HN1 overexpression. The p-H3^(S10)^ levels decreased significantly upon HN1 overexpression, and this decrease was more prominent after 4 h (Figure 4A). Although the efficiency of transfection was quite low as it was performed during the Nocodazole block (total 40-hr treatment), a small fraction of HN1-overexpressing cells remained delayed at the S phase, where a significant population of mitotic cells reached G1 after an early exit from mitosis that was shown by flow cytometry (Figure 4B). Since HN1 was overexpressed after the Nocodazole block (where both control and HN1 overexpressed cells are expected to be at prometaphase), HN1-transfected cells exhibited a higher G1 population at 1 h after Nocodazole release in comparison to the control cells. However, the G1 cells progressed to the S phase faster in HN1 overexpressed samples as compared to control cells in the 4- and 8-h samples (Figure 4B). The raw data for PI staining for this experiment is also shown (Appendix A, Appendix A). The transient transfection in prolonged Nocodazole treatments is a drawback as it can interfere with the cellular phenotypes and transfection efficiency; Doxycycline induction in the Tet-ON system could help overcome the drawback. When HN1 was induced after the Nocodazole block with a short burst (6 h) of Doxycycline, Cdt1 levels (a G1 marker) increased faster than control cells (Figure 4C), implying that HN1-induced cells exited mitosis relatively faster as compared to control cells, which was confirmed in the flow cytometry analysis from the same treatments (Figure 4D).

### 3.3. HN1 Associates with Cell Cycle Regulators and Modulates Cyclin B1 Levels

Because the levels of Cyclin B1 were strongly downregulated by HN1 overexpression (Figure 3A and Figure 4A) as observed in cells transfected with HM-HN1, we examined the putative link between HN1 and Cyclin B1 degradation. PC3 cells synchronized using Nocodazole and control cells were subjected to immunoprecipitation using control IgG and anti-Cyclin B1. It was observed that HN1, although lower in Nocodazole arrested population as compared to control cells, was physically associated with Cyclin B1 (Figure 5A). The results imply that Cyclin B1 degradation might be regulated through HN1 expression.

Cyclin B1 degradation occurs via proteasomal degradation [27] in HN1 overexpressed cells through proteasome machinery. Therefore, we transfected PC3 cells with HN1-overexpressing and/or an empty vector. The cells were also treated with the proteasome inhibitor MG132 or with DMSO. Interestingly, HN1 overexpression-mediated substantial decrease in Cyclin B1 level was reversed with MG132 treatment (Figure 5B). Since Cyclin B1 degradation depends on increased ubiquitination, HN1 could potentially contribute to Cyclin B1 degradation through its ubiquitination. To test this hypothesis, lysates from HN1-transfected PC3 cells were used for determining the Cyclin B1 ubiquitination level (Figure 5C). Our results revealed that HN1 overexpression is associated with increased rates of Cyclin B1 ubiquitination. The effect of HN1 overexpression using the Tet-ON system is also shown (Figure 5D,E). With the increase in the duration of Doxycycline treatment, HN1 ectopic levels increased; meanwhile, Cyclin B1 protein levels decreased. Moreover, HN1 localized with Cyclin B1 in the G2 and prometaphase phase as observed in immunofluorescence microscopy (Figure 5F). The immunofluorescence data complies with the immunoprecipitation data (Figure 5A), where cells synchronized in post-G2 using the Nocodazole block show specific interaction of Cyclin B1 with HN1.

The APC/C complex, the major player involved in Cyclin B1 degradation, starts to get activated right at the metaphase to anaphase transition [28]. Since Cdh1 is one of the co-factors of the APC/C and plays an important role in the degradation of Cyclin B1 [29], we immunoprecipitated Cdh1 and observed its association with ectopic HN1 (Figure 6B). Furthermore, the other co-factor of APC/C, Cdc20, is also involved in Cyclin B1 degradation, whereas the HN1 overexpression influenced solely Cdh1 (both temporal activators of APC/C) [14], but not Cdc20, expression (Figure 6(A1)). The lysates were also treated with λ-phosphatase to observe any impact of HN1 overexpression on phosphorylations of either of these co-factors. We observed that the HN1 overexpression led to the stabilization of Cdh1 (Figure 6(A2)). Moreover, in DU-145 cells, HN1 overexpression also led to the stabilization of Cdh1, while Cdh1 overexpression decreased HN1 ectopic levels (Figure 6C). Additionally, HN1 and Cdh1 overexpressed samples showed a distinct interaction of HN1 with Cdh1 in both cell lines tested (Figure 6B,C). Therefore, the data suggest that the HN1 directly associates with Cdh1 for its stabilization regardless of its phosphorylations. Since APC/C-Cdh1 directly acts to increase Cyclin B1 ubiquitination, HN1 could potentially impact Cyclin B1 degradation via the ubiquitination pathway. Taken together, these results support the conclusion that ectopic HN1 expression at prometaphase and mitosis allows cells to precede mitosis earlier [30,31]. This result might be the consequence of unexpected Cyclin B1 degradation by the overexpression of HN1 at pro-metaphase or post-G2 phases of the cell cycle. These results indicate that HN1 is involved in the regulation of cell cycle mediators.

HN1 depletion using siRNA was also performed to detect the levels of Cdk1, and Emi1 (a negative regulator of APC/C activity [32]) in PC3 cells in either asynchronized (control) or Nocodazole arrested cells released into G1 for up to 4 h. HN1 silencing led to the stabilization of Cdk1 and Emi1 (Figure 7).

### 3.4. HN1 Is Directly Involved in The Regulation of S-Phase Dynamics

Since HN1 overexpression led to S-phase accumulation, further experiments to validate the previous findings were conducted. HN1 was fused with the Venus fluorescence tag at C-terminus in the Tetracycline inducible system and a stable PC3 cell line was generated after viral transduction. The system was validated using fluorescence microscopy (Figure 8A). PC3 cells stabilized with the HN1-Venus inducible system were further investigated for anti-BrdU staining after treatment with BrdU for two hours before cell fixation for immunofluorescence assay (Figure 8(B1)). The HN1-Venus-expressing cells showed a higher percentage of BrdU staining, which demonstrated that HN1 functions in the progression of the S-phase (Figure 8(B2)). Moreover, PC3 cells stabilized with the HM-HN1 inducible system were also fixed and stained with anti-BrdU and PI for flow cytometry analysis (Figure 8(C1)). The raw data for flow cytometric analysis are shown in Appendix A. The number of cells incorporating BrdU increased in HN1 overexpression samples (Figure 8(C2)). Therefore, we suggest that HN1 directly plays a role in the “early progression” of the S-phase as native HN1 levels are also higher in S cells, whereas putatively, a certain threshold is achieved to retain the cells in the S phase onset (Figure 3B,D).

DU-145 cells were stabilized with the HM-HN1 inducible system, and the cells were processed for cellular fractionations for cytosolic, nuclear, and chromatin separation (Figure 9). Western blotting confirmed HN1 induction upon Doxycycline treatment in cytosolic and nuclear fractions of DU-145 cells. Here, we observed that the replication origin licensing factor, Cdt1 levels, increased in the cytoplasm in HN1 overexpressed samples with a decrease in chromatin fractions. In the nuclear fraction, Cdt1 levels remained unchanged upon HN1 overexpression. Since the HN1 phospho-form is observed higher in the nuclear fraction, the data suggested that the S phase increase in HN1 levels has to be kept at a certain threshold to keep the Cdt1 function normal, whereas the increase in HN1 expression leads to Cdt1 retention in the cytoplasm. In PC3 cells, HN1 itself was not observed in the chromatin fraction of the lysates and therefore has not been shown.

## 4. Discussion

HN1 was discovered in 1997 [33], and due to higher expression in murine hematological and neurological tissues, it was named HN1. Later, a study was conducted to find the homologs of HN1 and it was cloned from the human cDNA library [34]. In that study by Zhou et al. in 2004, it was claimed that cells stably expressing HN1 led to S-phase retardation [34]. In another study, HN1 depletion led to a p21 increase and G1 arrest in B16.F10 cells [35]. HN1 depletion also resulted in G1 arrest and increased p21 levels in Hepatocellular Carcinoma cells [36]. Recently, HN1 was investigated in light of transcriptome data of Prostate adenocarcinoma, where HN1 co-expressed genes included two distinct nodes of S-phase and mitotic regulators [13]. In that study, HN1 expression was observed as higher in Prostate cancer samples as compared to normal adjacent Prostate tissues from the patients. Therefore, this study focused on further dissecting the role of HN1 in Prostate cancer cell lines. We demonstrated that HN1 levels are higher in cancer cells as compared to normal epithelial cells for both the Prostate and Breast (Appendix A).

HN1 levels fluctuated in the cell cycle, remaining enriched in the S phase and went down after the S phase until the end of G1 with a transition into a phosphorylated form only appearing in mitosis (Figure 1) with potential upstream kinases as GSK3β and Cdk1 (Figure 1). HN1 has been shown as a regulator of the GSK3β signaling pathway previously [9,37]; the inhibition of p-HN1 upon Cdk1i and pan-Cdki treatments implies that HN1 is involved in pathways regulated via Cyclin-dependent kinases (CDKs). The CDK1 regulation of HN1 should be explored further.

Robust checkpoint regulators including Cyclins, Cyclin-dependent kinases, and their inhibitors control the cell cycle [38]. Cell cycle progression is interrupted when one of these factors is abrogated. A loss of checkpoint regulator function usually results in reduced control of cellular growth and the perpetuation of genomic defects [39]. Cyclin B1, known as mitotic Cyclin, contributes to the G2/M checkpoint transition by activating Cdk1, ensures the appropriate cellular responses to genomic insults or abnormalities, and accordingly guarantees the continuation of faithful cell division [40,41]. The expression of Cyclin B1 begins at the late S or early G2 phase, and the protein accumulates until it reaches its highest level at the metaphase to anaphase transition checkpoint known as the Spindle Assembly Checkpoint (SAC) [42,43]. Cyclin B1 forms a complex with Cdk1 during G2 to control the G2/M transition and SAC in cell division. Because the APC/C mediated proteasomal degradation of Cyclin B1 occurs during the metaphase/anaphase transition and continues until the S phase of the subsequent cell cycle, the factors involved in the timing of Cyclin B1 degradation in the metaphase/anaphase transition are important cell cycle mediators and complete mechanisms also remain unclear [21]. HN1 overexpression regardless of cell cycle phase led to Cyclin B1 decrease (Figure 3A and Figure 4A). HN1 was immunoprecipitated with Cyclin B1 in Nocodazole arrested cells and HN1 overexpression led to increased ubiquitination of Cyclin B1 as well. Therefore, HN1 is directly involved in regulating the cell cycle, which was observed in altered mitotic exit dynamics of HN1 overexpression after Nocodazole arrest with two different approaches (Figure 3 and Figure 4).

Nocodazole-arrested cells have higher cyclin B1 levels as compared to non-synchronized cells. Cyclin B1 levels start to manifest in the G2 phase and are enriched until mitosis [44]. Therefore, to understand whether cyclin B1 decrease was due to cell cycle arrest upon HN1 overexpression or if there is a direct link between HN1 and cyclin B1 degradation dynamics, immunoprecipitation was performed for control and Nocodazole arrested cells and HN1 was observed to interact with cyclin B1, which shows that cyclin B1 degradation could be the result of HN1 overexpression and seen in increased ubiquitination levels of cyclin B1. Normally, cyclin B1 is degraded in metaphase until late S, where APC/C remains active [30]. HN1 overexpression changes the cellular phenotype upon interacting with centrosomes and other regulators such as GSK3β [11] along with cyclin B1 and Cdh1; the overall levels of cyclin B1 ubiquitination levels present one of the forms of this changed phenotype. Moreover, cyclin B1 degradation was also viewed as a function of proteasome activity in MG132 treatments with or without HN1 overexpression and the results significantly lead to the conclusion that HN1 is directly linked with Cyclin B1 degradation as a direct interaction and enhanced APC/C activity as a mechanism. Recently, it has been shown that HN1 silencing sensitizes Prostate cancer cells to chemotherapeutic agents such as 2-Methoxyestradiol and Docetaxel via Cyclin B1 regulation [19]. The same study showed the inverse correlation of the levels of HN1 and Cyclin B1. HN1 acts as an anti-apoptotic protein in Prostate cancer and therefore, should be considered for further investigations related to its mechanisms.

The interaction of APC/C co-factor Cdh1 with HN1 is also interesting, as APC/C-Cdh1 is known to degrade mitotic Cyclin B1 and other mitosis-related factors such as PLK1 and Aurora A [28]. Moreover, HN1 overexpression stabilized the Cdh1 levels, implying that HN1 could have a role in enhancing APC/C activity via Cdh1, which contributes to Cyclin B1 degradation [16,27]. Cdh1 overexpression decreases the accumulation of Cyclin B1 in different contexts [45,46,47]. Since the FUCCI system provides a unique tool to decipher APC/C-Cdh1 activity in live cells [32,48,49], the effect of HN1 overexpression on APC/C was observed in FUCCI-PC3 cells where the index of APC/C-Cdh1 activity was higher in cells stably expressing HN1 (unpublished data). Therefore, mechanistic insight is developed regarding the role of HN1 in the cell cycle as a result of this study, where HN1 interacts with Cdh1 for its stabilization and increasing APC/C-Cdh1 activity to increase the ubiquitination and subsequent degradation of Cyclin B1. The proposition is further strengthened with HN1 depletion causing accumulation of Emi1, a major inhibitor of APC/C-Cdh1 activity (Figure 7) [32]. The evidence that HN1 levels decrease while approaching mitosis after the S phase, and HN1 overexpression augmenting early exit from mitosis, implies that HN1 downregulation is required for normal mitosis, where the chromosomal congression and bipolar spindle assembly occur [50,51]. Since previously, HN1 has been shown to regulate centrosome and microtubule organizations in prostate cancer cells [13]; it is therefore concluded that HN1 is a cell cycle regulatory protein.

HN1 is upregulated in Hepatocellular carcinoma [36,37,52,53], promotes anaplastic thyroid carcinoma [54], and metastases in Cervical carcinoma [55]. HN1 expression has been associated with MYC, where HN1 overexpression was linked with increased migration and invasiveness in Breast cancer [56]. HN1 expression has been reported to be higher in tumor samples as compared to normal tissues of the prostate and breast [13,56]. Moreover, HN1 was observed as one of the ten most highly differentially expressed genes (up-regulated) in Pancreatic ductal adenocarcinoma as compared to normal tissues [57]. HN1 expression has also been associated with the worst overall survival in Breast cancer patients [58]. Therefore, exploring HN1 for its interactions and cellular mechanisms, and pathways, is important in the context of cancer as it is a potential proto-oncogene involved in promoting the carcinogenic potentials of cancer cells. Here, we demonstrated that HN1 overexpression led to the progression of Prostate cancer cells to the S-phase when stable overexpression in BrdU stainings using the Tetracycline inducible system (Figure 8) and transient overexpression studies were conducted (Figure 3 and Figure 4). Cdt1 is a replication licensing factor, which remains bound to chromatin in G1 and is unloaded from chromatin upon initiation of replication [59]. HN1 overexpression increased cytoplasmic retention of Cdt1 (Figure 9), implicating the mechanism of early S-phase entry and accumulation of HN1 overexpressed cells in the S-phase of the cell cycle [60]. Briefly, this study brought forth the novel mechanism of the HN1/Cdh1/Cyclin B1 axis for the regulation of mitosis and unloading of Cdt1 from chromatin for G1/S transition. The S-phase entry and the role of HN1 in replication control machinery are yet to be defined and need further mechanistic studies.

## 5. Conclusions

In conclusion, we provided a novel insight into the function of HN1 in regulating the cell cycle, evidenced in the temporal accumulation and downregulation of HN1 levels in the cell cycle. HN1 overexpression using transient and stable overexpression systems showed that HN1 is involved in G1/S and M/G1 transitions. It associated with Cyclin B1 and downregulated it, and on the other hand, associated with Cdh1 and stabilized it. The increased ubiquitination of Cyclin B1, and reversal of downregulation upon the proteasome inhibitor in HN1 overexpressed samples as compared to controls, suggest that HN1 contributes to Cyclin B1 degradation. The increased accumulation of HN1 overexpressed cells in the S-phase was due to increased retention of replication licensing factor Cdt1 in the cytoplasm and its reduced loading onto chromatin. Moreover, HN1 is inhibited upon Cdk1 inhibition, also providing a unique opportunity to study HN1 further in the context of the cell cycle. Studies employing HN1 phospho-mutants along with investigation of HN1 interactions with further key elements of cell cycle regulation machinery can help bring light to the role of HN1 in mechanisms for G1 to S entry and mitotic exit.

## Figures and Tables

**Figure 1 biology-12-00189-f001:**
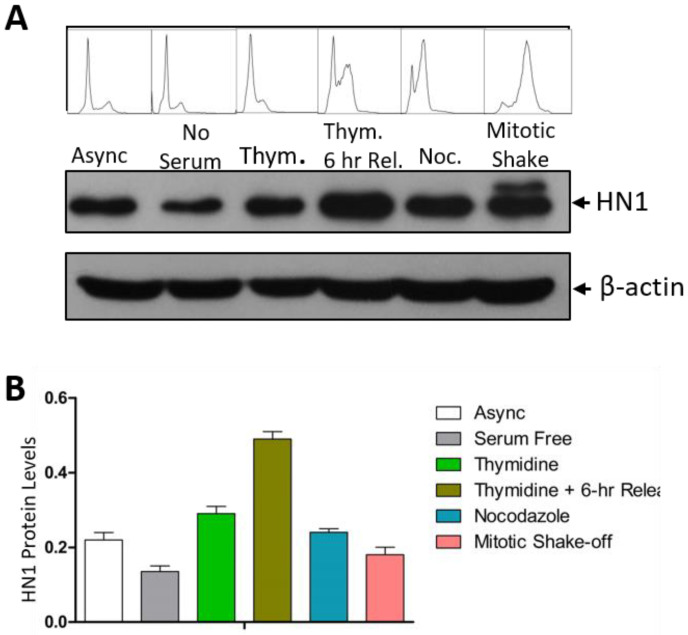
**HN1 fluctuates during the cell cycle:** (**A**) PC-3 cells synchronized by different approaches were subjected to western blotting (serum-free (G1), Thymidine block (G1/S), Thymidine block and release (S), Nocodazole block (post-G2), and mitotic shake-off (M)). HN1 protein levels fluctuated in different phases of the cell cycle. Cell cycle distribution profiles obtained by PI staining and flow cytometry analysis are also shown above each sample. The β-actin antibody was used as the loading control in the experiment. (**B**) Densitometric analysis portraying relative HN1 protein levels normalized to β-actin is shown to quantify HN1 in cell cycle phases.

**Figure 2 biology-12-00189-f002:**
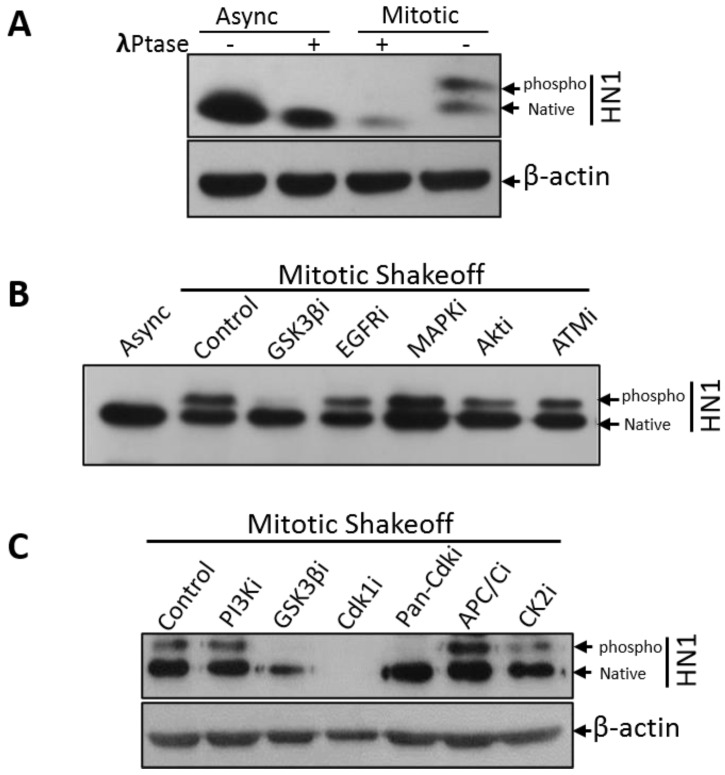
**HN1 is phosphorylated in mitosis:** (**A**) PC3 controls and mitotic lysates were treated with λ-phosphatase and immunoblot suggests that HN1 is phosphorylated in mitosis. The β-actin antibody was used as the loading control in the experiment. (**B**) Inhibitors of GSK3β (SB216763, 10 nM), EGFR (AG556, 10 μM), MAPK (PD98059, 1 μM), AKT pathway (Wortmannin, 100 nM), and ATM proteins (CAS 587871-26-9, 0.5 μM) were administered to PC3 cells (4 h before harvesting cells) after synchronization with Nocodazole in the prometaphase (or post-G2) phase and released into mitosis for 1 h where cells were harvested using the mitotic shake-off method. HN1 was labeled using an anti-HN1 antibody to determine the effect of various kinase inhibitors on the phospho-form of HN1. Only GSK3β inhibitor treatment led to the disappearance of p-HN1. (**C**) In PC3 cells, mitotic shake-off was performed to collect mitotic populations, and cells were treated with different kinase inhibitors such as PI3K inhibitor (LY294002, 25 μM), GSK3β inhibitor (SB216763, 10 nM), CDK1 inhibitor (RO-3306, 10 μM), pan-CDK inhibitor (Flavopiridol, 500 nM), and CK1 inhibitor (TBCA, 0.5 μM), along with APC/C inhibitor TAME (200 μM). Protein lysates were subjected to western blotting whereupon anti-HN1 and anti-β-actin levels were measured. GSK3β inhibitor, Cdk1i, and pan-Cdki led to the inhibition of p-HN1, with Cdk1i also leading to a dramatic inhibition of native HN1.

**Figure 3 biology-12-00189-f003:**
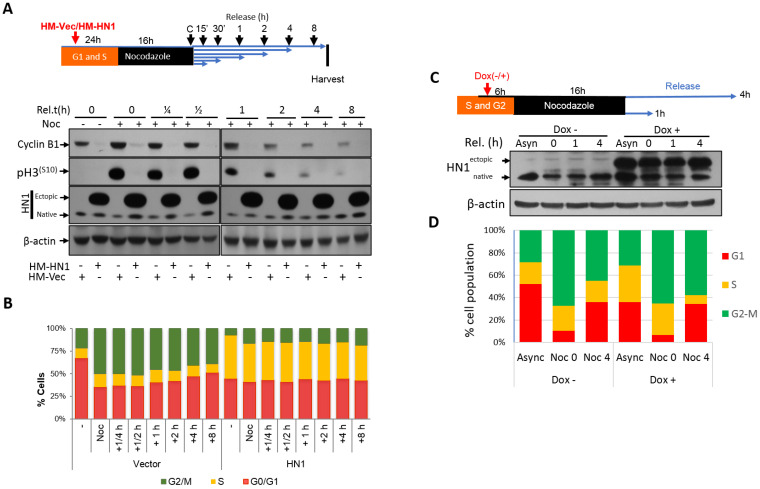
**HN1 overexpression before Nocodazole Block (post-G2) using the transient transfection and Doxycycline inducible system:** (**A**) The design of the first experimental approach is given where HN1 was overexpressed using HM-HN1 and the control vector (HM-Vec) was used as the control. Cyclin B1 and p-H3^(S10)^ levels decreased in HN1-overexpressing cells treated with Nocodazole and released at different times. (**B**) After 16 h of Nocodazole treatment, cells were released from the Nocodazole block and were analyzed using flow cytometry 0, 15, and 30 min and at 1, 4, and 8 h later. When HN1 was ectopically expressed before the Nocodazole block, a considerable number of cells accumulated in the S phase as shown in the yellow populations. (**C**) PC3 cells transduced with pCW57-HM-HN1 viruses were selected with puromycin and the stabilized cell line was generated. Furthermore, to determine the impact of HN1 induction using Doxycycline before Nocodazole Block (post-G2), cells were given Doxycycline 6 h before Nocodazole. The cells were kept in Nocodazole for 16 h and released at indicated times. HN1 induction was validated with immunoblotting using an anti-HN1 antibody, and anti-β-actin was used as a loading control. (**D**) The samples (Async, Noc 0 h, and Noc 4 h indicating release from Nocodazole block (post-G2) to exit from mitosis and entry into G1) were subjected to cell fixation and PI staining followed by flow cytometry.

**Figure 4 biology-12-00189-f004:**
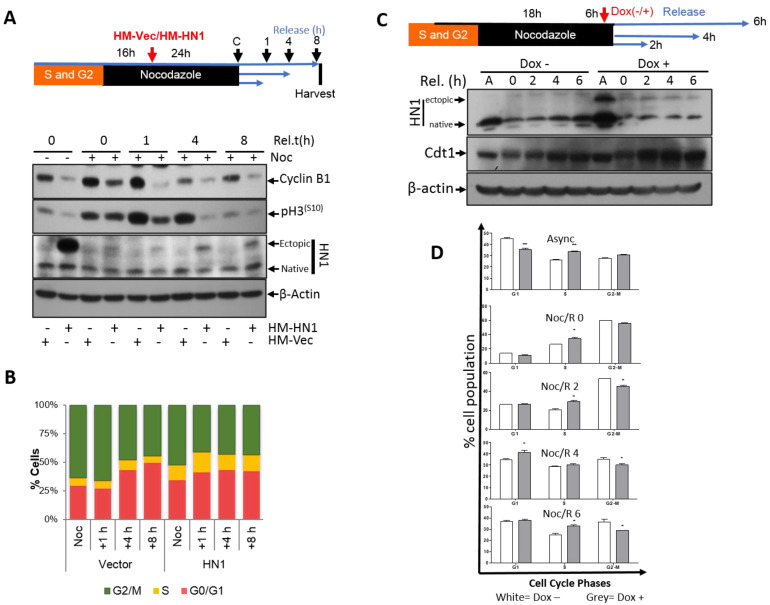
**HN1 overexpression after Nocodazole Block (post-G2) using transient transfection and the Doxycycline inducible system**. (**A**) In the second experimental approach, HN1 was ectopically expressed during Nocodazole arrest, where cells were allowed to reach prometaphase (or post-G2) first with Nocodazole treatment for 16 h and then HN1 was overexpressed using HM-HN1 plasmid and released into G1 after release from the Nocodazole block for up to 8 h. Again, in control cells, p-H3^(S10)^ levels decreased along with Cyclin B1. (**B**) The flow cytometry analysis showed that HN1 overexpressed cells left mitosis early as observed in 1 h release samples, and afterward, S phase populations increased in HN1 overexpressed cells. Therefore, it can be deduced that HN1 plays an important role in the temporal mitotic transition to G1 cells. (**C**) pCW57-HM-HN1 stable PC3 cells were synchronized by applying Nocodazole at a final concentration of 165 nM and released in normal fresh media at indicated periods. The Doxycycline treatment period was kept the same at 6 h in all samples. For instance, the cells treated with Nocodazole and not released from the Nocodazole block were treated with Doxycycline in the presence of Nocodazole for 6 h. Similarly, the cells released from the Nocodazole block for 2 h remained in Doxycycline and Nocodazole for 4 h and after adding fresh media for release, Doxycycline was added for the duration of the release, i.e., 2 h (hence total Doxycycline treatment is 6 h). The protein lysates were subjected to western blotting for anti-HN1, anti-Cdt1, and anti-β-actin antibodies. The upper bands in the HN1 lane show ectopic expression of HM-HN1 in Doxycycline-treated samples. (**D**) The cells stained with PI were subjected to flow cytometry analysis to reveal the cell cycle distribution analysis. The white bars represent cells not treated with Doxycycline and the grey bars represent cells treated with Doxycycline for the induction of ectopic HN1 expression. The statistical analysis of the data shows some significant differences that are illustrated by (*) signs with a *p*-value less than 0.05.

**Figure 5 biology-12-00189-f005:**
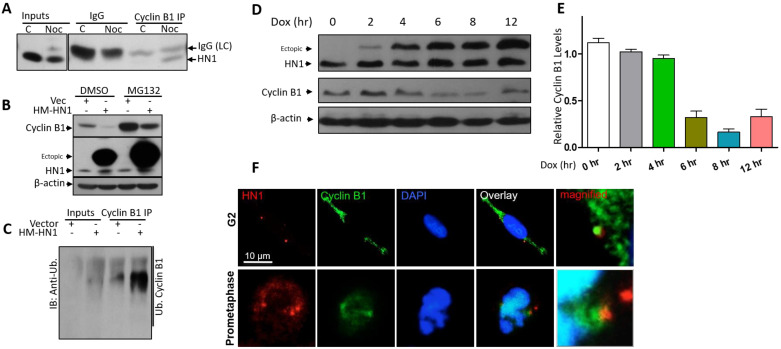
**HN1 is directly linked with Cyclin B1 degradation**: (**A**) To determine the interaction of HN1 with Cyclin B1, anti-Cyclin B1 was used for immunoprecipitation in control and Nocodazole treated (post-G2 arrested) PC3 cells and subjected to immunoblotting using an anti-HN1 antibody. (**B**) HN1 was overexpressed in PC3 cells and treated with either DMSO or the proteasome inhibitor MG132 to observe the decrease in Cyclin B1 expression caused by HN1 overexpression due to proteasome machinery. (**C**) The cell lysates from vector- or HN1-transfected cells were subjected to immunoprecipitation with an anti-Cyclin B1 antibody and were probed with an anti-ubiquitin antibody. Cyclin B1 ubiquitination increased subsequently to HN1 overexpression. (**D**) PC3-pCW57-HM-HN1 cells were given Doxycycline for indicated periods (2, 4, 6, 8, 12 h) and protein lysates were subjected to immunoblotting using anti-HN1, anti-Cyclin B1, and anti-β-actin antibodies. HN1 ectopic expression increased temporally with Doxycycline exposure and Cyclin B1 bands showed decreased levels. (**E**) Densitometric analysis of relative Cyclin B1 intensities from (**D**) is shown. (**F**) Images of PC3 cells at G2 and prometaphase are shown where HN1 is labeled with red fluorescence and Cyclin B1 is labeled with green fluorescence exhibiting secondary antibodies. Nuclei are counterstained with DAPI (10 μm scale bar is shown).

**Figure 6 biology-12-00189-f006:**
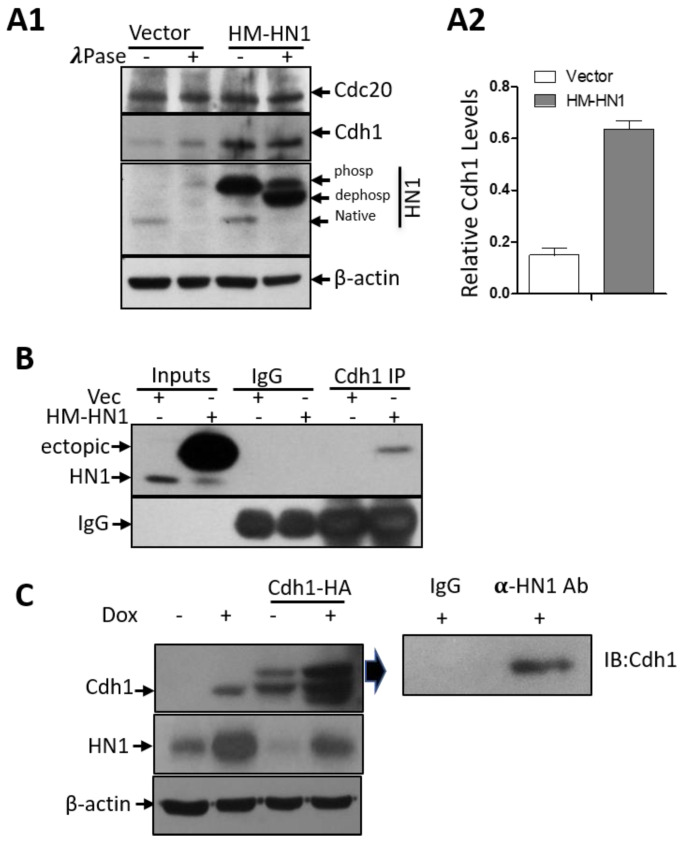
**HN1 stabilizes and interacts with Cdh1.** (**A1**) The effect of HN1 overexpression on phosphorylations of Cdc20 and Cdh1 was queried using λ-phosphatase assay on protein lysates collected after vector and HM-HN1 transfections in PC3 cells. Anti-β-actin was used as the loading control of the experiment. (**A2**) The Cdh1 protein levels normalized to β-actin are shown. (**B**) Since Cdh1 is the co-factor of APC/C, which was stabilized upon HN1 overexpression, Vector and HM-HN1 transfected samples were subjected to immunoprecipitation using the anti-Cdh1 antibody and immunoblotted with the anti-HN1 antibody. In all immunoprecipitation assays, control IgG antibodies from the same host as IP antibodies were utilized. (**C**) DU-145 cells stabilized with pCW57-HM-HN1 viruses were induced with Doxycycline (1 ug/mL) and also transfected with Cdh1-HA plasmid (Addgene No. 11596) to observe changes in Cdh1 and HN1 levels with immunoblotting. HN1 overexpression with Doxycycline led to the stabilization of Cdh1 levels, while Cdh1 overexpression decreased HN1 levels. The sample with both Cdh1 and HN1 overexpression was subjected to immunoprecipitation using an anti-HN1 antibody crosslinked with Dynabeads G and immunoblotted with an anti-Cdh1 antibody. HN1 interacted with Cdh1 in this sample.

**Figure 7 biology-12-00189-f007:**
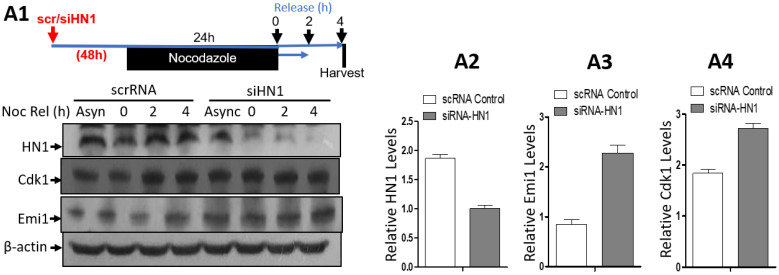
**HN1 depletion using siRNA leads to the stabilization of CDK1 and Emi1.** (**A1**) PC3 cells were either given scrambled control RNA or siRNA targeting HN1 (siHN1), treated with Nocodazole for 18 h, and released into G1 for up to 4 h by replacing Nocodazole with fresh media. The cells were subjected to protein isolation and subsequent western blotting, where the relative levels of HN1, Cdk1, and Emi1 were measured using immunoblotting, and anti-β-actin was used as the loading control of the experiment. The relative HN1 levels (**A2**), Emi1 levels (**A3**), and Cdk1 levels (**A4**) are measured using densitometry analysis normalized to β-actin.

**Figure 8 biology-12-00189-f008:**
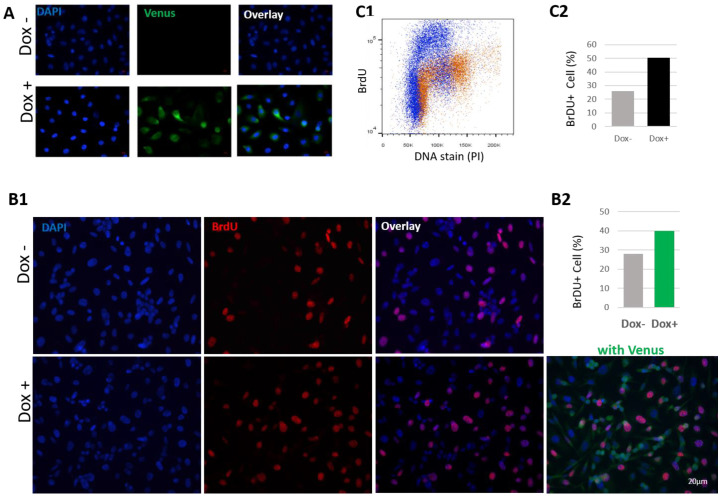
**HN1 is directly associated with the progression of the S phase.** (**A**) PC3 cells transduced with pCW57-HN1-Venus viruses were selected with puromycin (2 μg/mL), plated on coverslips and induced with Doxycycline for 2 days, fixed with 4% paraformaldehyde, and analyzed using fluorescence microscopy to validate the expression of HN1-Venus. DAPI was used to stain nuclei. (**B1**) PC3 cells stabilized with pCW57-HN1-Venus were induced with Doxycycline for 2 days and BrdU was applied 2 h before cell fixation. Anti-BrdU staining was performed (secondary anti-mouse Alexa Fluor-594) to observe cells actively undergoing DNA synthesis, where DAPI was used for nuclei staining. (**B2**) BrdU positive cells counted (from B1) were graphed and it was observed that HN1-Venus expressing cells produced significantly higher amounts of BrdU positive cells. The mean number of BrdU+ cells in Dox− samples were 36/129 nuclei and in Dox+ samples were 57/143. The graph indicates the mean percentages of BrdU+ cells. (**C1**) PC3 cells stabilized with pCW57-HM-HN1 viruses were subjected to Doxycycline induction of ectopic HN1 expression and BrdU treatment (as performed in (**B**) and cells were fixed in ethanol (70% in PBS)). After PI and anti-BrdU staining, cells were analyzed in a flow cytometer, and it was observed that HN1 overexpressing cells exhibited higher levels of BrdU-stained cells. The representative image processed with the population-gating procedure is shown. (**C2**) The mean number of BrdU+ cells are compared in the form of a graph, where HN1 overexpression increased BrdU incorporation in PC3 cells.

**Figure 9 biology-12-00189-f009:**
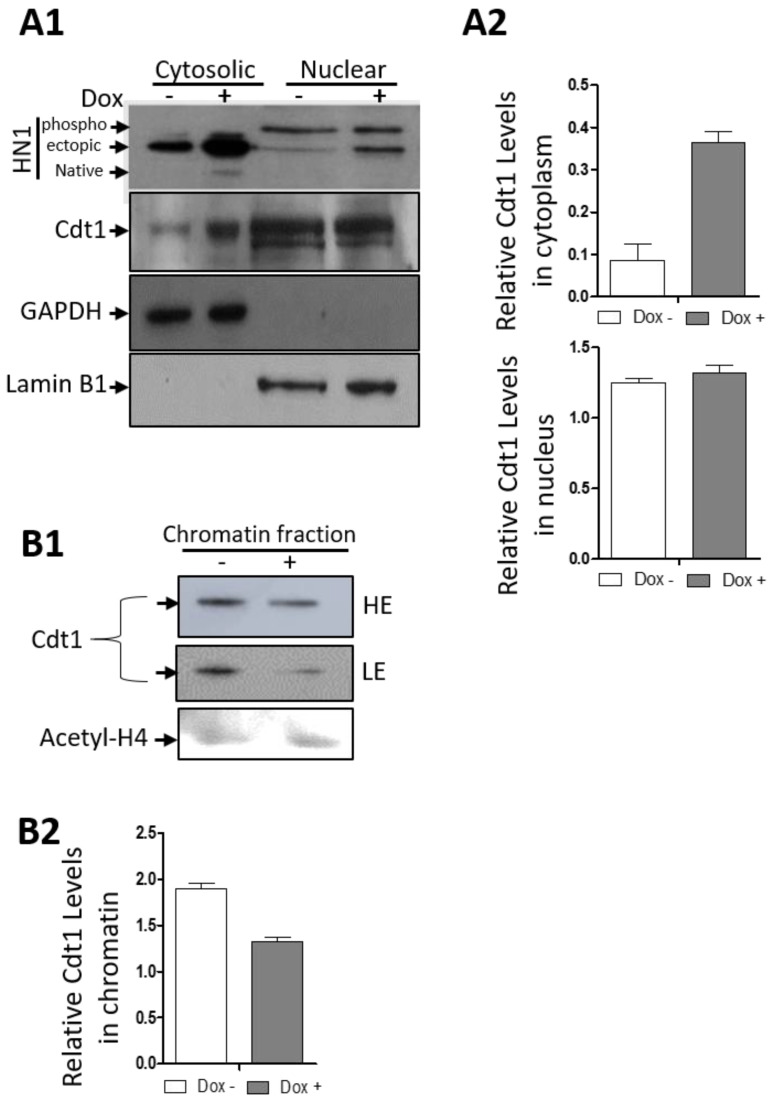
**HN1 overexpression leads to cytoplasmic retention of Cdt1.** (**A1**) The DU-145-pCW57-HM-HN1 cells were induced with Doxycycline and subjected to cytosolic, nuclear, and chromatin fractionation. The ectopic HN1 expression was also observed in cells not treated with Doxycycline and therefore, represents the leaky ectopic expression of HN1. HN1 was observed in all nuclear and cytosolic fractions. For cytosolic and nuclear fractions, other antibodies checked for immunoblotting included anti-Cdt1, anti-GAPDH, and anti-Lamin B1. Anti-GAPDH was used as a cytosolic control and anti-Lamin B1 was used as a nuclear control. (**A2**) Cdt1 levels in cytoplasm normalized to GAPDH and in nucleus normalized to Lamin B1 are shown. (**B1**) For chromatin samples, anti-Cdt1 and anti-acetylated H4 were used. (**B2**) The Cdt1 levels in chromatin normalized to Acetyl-H4 are shown. (HE = High Exposure, LE = Low Exposure).

## Data Availability

All data pertaining to this publication can be requested from the corresponding author and can be made readily available.

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
