# Peer review of "HN1 Is Enriched in the S-Phase, Phosphorylated in Mitosis, and Contributes to Cyclin B1 Degradation in Prostate Cancer Cells"

_biology, 2023, doi:10.3390/biology12020189_

Round 1

Reviewer 1 Report

Biology
Manuscript ID: biology-2141533
HN1 is Enriched in S-phase, Phosphorylated in Mitosis, and Contributes to Cyclin B1
Degradation in Prostate Cancer Cells
By Aadil Javed, Gülseren Özduman, Lokman Varışlı, Bilge Esin Öztürk, Kemal Sami Korkmaz
The authors studied the role of the Hematological and neurological expressed 1 (HN1) in cell
cycle in the context of prostate cancer. They reported that HN1 is tightly regulated during the
cell cycle and described a specific phosphorylated form in mitosis. HN1 overexpression leads
to S-phase accumulation and causes early exit after post-G2 overexpression. At the molecular
level, they found that HN1 co-immunoprecipitates with cyclin B and modulates its stability
through ubiquitination. They also documented the HN1 involvement in the regulation of Sphase.
Finally, these studies provide a novel insight into the function of HN1 in regulating the
cell cycle.
Although these data are innovative and with potential involvement in understanding its role in
the modulation of cell cycle progression at the S-phase and mitotic exit, some key points
deserve to be improved before considering for publication in Biology.
Point-by-point comments:
Figure 5: HN1 is directly linked with Cyclin B1 degradation.
5A. The immunoprecipitations must be performed with the same antibody comparing
transfected cells with siRNA CTL versus siRNA against cyclin B1. The reverse
immunoprecipitation can be included as well ie immunoprecipitation with HN1 antibody and
western blotting with cyclin B1 antibody.
Authors should provide an image of the western blotting performed with cyclin B1 and b-actin
antibody.
5C. Authors should include more controls: western blot with antibody against cyclin B, HN1
and b-actin. Without these informations, we cannot interpret the experiments. Did authors
immunoprecipitate same cyclin B1 level?
Figure 6: HN1 stabilizes and interacts with Cdh1.
6B. Same comment as 5A.
Please could you provide western blot for Cdh1, b-actin. It seems that the quantity of antibody
used for the immunoprecipitation is not the same between IgG and Cdh1.
6C. the quality of the image is not enough.
Please include an image with HA antibody.
The exposure time is not adequate for b -actin.
Please perform reverse immunoprecipitation with antibody against HN1 in DU-145 cells
transfected with siRNA CTL and siRNA HN1.
Figure 7: HN1 depletion using siRNA leads to the stabilization of CDK1 and Emi1.
Authors should improve the quality of the image presented.
Please could you quantify 3 independent experiments and provide the graph?
Figure 8: HN1 is directly associated with the progression of the S-phase.
6C and D. I could not observe the BrdU-positive cells (around 35 %) on the flow cytometry
panel for the dox-negative cells. Please clarified this point.
6B. Please could you quantify the fluorescence and provide the graph?
Please include scale bare.
Figure 9: HN1 overexpression leads to cytoplasmic retention of Cdt1.
9A. “Cdt1 levels increased in the cytoplasm in HN1 overexpressed samples with a decrease
in nuclear and chromatin fractions…..” please provide quantification of 3 independent
experiments. I could not see this decrease on the image provided. Please authors should
increase the quality of this image.
9B. Controls are missing for interpretation of the experiment.
General comment:
The manuscript should be proofread for the English and editing … example line 272…

Author Response

The authors studied the role of the Hematological and neurological expressed 1 (HN1) in cell cycle in the context of prostate cancer. They reported that HN1 is tightly regulated during the cell cycle and described a specific phosphorylated form in mitosis. HN1 overexpression leads to S-phase accumulation and causes early exit after post-G2 overexpression. At the molecular level, they found that HN1 co-immunoprecipitates with cyclin B and modulates its stability through ubiquitination. They also documented the HN1 involvement in the regulation of Sphase. Finally, these studies provide a novel insight into the function of HN1 in regulating the cell cycle. Although these data are innovative and with potential involvement in understanding its role in the modulation of cell cycle progression at the S-phase and mitotic exit, some key points deserve to be improved before considering for publication in Biology.

Response: We appreciate and acknowledge the thorough analysis and positive comments for our manuscript by the reviewer. The suggestions made by the reviewer are valid and have been incorporated into the revised manuscript, which has improved the quality of the presentation of our paper.

Point-by-point comments:

Figure 5: HN1 is directly linked with Cyclin B1 degradation. 5A. The immunoprecipitations must be performed with the same antibody comparing transfected cells with siRNA CTL versus siRNA against cyclin B1. The reverse immunoprecipitation can be included as well ie immunoprecipitation with HN1 antibody and western blotting with cyclin B1 antibody. Authors should provide an image of the western blotting performed with cyclin B1 and b-actin antibody. 5C. Authors should include more controls: western blot with antibody against cyclin B, HN1 and b-actin. Without these informations, we cannot interpret the experiments. Did authors immunoprecipitate same cyclin B1 level?

Response: The western blotting performed with Doxycycline inducible HN1 cells (PC3-pCW57-HMHN1) for indicated periods (0, 2, 4, 6, 8, 12 hrs) has been added to the manuscript. Briefly, Cyclin B1 levels decrease temporally with induction of HN1, further strengthening the conclusions of the paper that HN1 is linked with Cyclin B1 degradation. The quantitated data has also been added. Immunoprecipitations performed with silenced proteins are not recommended as we have already shown that siRNA-HN1 transfections deplete HN1 and therefore if the protein is depleted, it will not precipitate with either Cyclin B1 or HN1 antibodies. The immunoprecipitations with siRNA transfected samples are performed when the effect on downstream actors is to be determined with the protein which gets stabilized upon the depletion of the gene of interest. Therefore, we opted to perform immunoprecipitation using control lysates and Nocodazole arrested lysates to showcase HN1 interaction with Cyclin B1. Moreover, it has been shown in the paper that HN1 overexpression reduces Cyclin B1 (Figure 2A (control vs HN1 overexpressed), 5B (control vs HN1 overexpressed (not treated with MG132)) and in native conditions HN1 interacts with Cyclin B1, therefore we opted to show HN1 co-localization with Cyclin B1 using immunofluorescence with cells synchronized in G2 and pro-metaphase phases of the cell cycle. The higher expression of Cyclin B1 upon HN1 depletion can be observed in one of our previous papers (Reference 9). The immunofluorescence and western blotting data have been incorporated in the revised manuscript. The suggestions from the reviewer have increased the quality of the manuscript in our opinion. Since HN1 overexpression downregulates Cyclin B1 levels and Ubiquitin antibody was used for western blotting to show ubiquitinated Cyclin B1 levels upon HN1 overexpression, which are clearly increased. The reviewer can look at reference 37, where the effect of HN1 overexpression on c-myc ubiquitination was shown. We performed a similar experiment and also ran the input with IP samples.

Figure 6: HN1 stabilizes and interacts with Cdh1. 6B. Same comment as 5A. Please could you provide western blot for Cdh1, b-actin. It seems that the quantity of antibody used for the immunoprecipitation is not the same between IgG and Cdh1. 6C. the quality of the image is not enough. Please include an image with HA antibody. The exposure time is not adequate for b -actin. Please perform reverse immunoprecipitation with antibody against HN1 in DU-145 cells transfected with siRNA CTL and siRNA HN1.

Response: The lysates not treated with lambda-phosphatase clearly show that the HN1 overexpression leads to the stabilization of Cdh1. Moreover, the quantitated data from western blots showing higher Cdh1 levels upon HM-HN1 transfection has been added to the revised paper. Furthermore, the effect of HN1 overexpression on Cdh1 levels and interaction are presented using two different overexpression systems. Plasmid overexpression primarily is hard to do in a higher number of cells, this is the reason the second system of stable overexpression using doxycycline-inducible expression was used to get a higher percentage of cells expressing stable HN1 expression. In both cases, Cdh1 interaction with HN1 is shown. In one case, the anti-Cdh1 antibody is used for immunoprecipitation, and anti-HN1 is used for immunoblotting, meanwhile, in another case, the anti-HN1 antibody is used for immunoprecipitation and anti-Cdh1 is used for immunoblotting. Therefore, the results of reciprocal immunoprecipitations clearly imply that HN1 interacts with Cdh1. Since immunoprecipitation is not principally quantitative based on different antibodies used for control and specific antibodies. The background or IgG precipitated may vary from experiment to experiment. This is also the reason, in cells stably expressing HN1 and also overexpressing Cdh1 were used for immunoprecipitating HN1 using magnetic dynabeads Protein G. The use of anti-HA is not required here as the direct interaction has been shown on 2 separate occasions already. Immunoblots with better image quality have been added to the manuscript. Immunoprecipitation in HN1 silenced samples is not recommended as western clearly shows HN1 is depleted therefore there is no point in showing the interaction of depleted protein. It is emphasized that the data clearly demonstrate that HN1 interacts with Cdh1 and stabilizes its levels when overexpressed.

Figure 7: HN1 depletion using siRNA leads to the stabilization of CDK1 and Emi1. Authors should improve the quality of the image presented. Please could you quantify 3 independent experiments and provide the graph?

Response: We appreciate the suggestion by the reviewer for improving the quality of the manuscript. The images have been improved and the graphs have been added. We thank the reviewer for the suggestion. The quantitation of the blots has provided an additional layer of clarity for the interpretation of the data.

Figure 8: HN1 is directly associated with the progression of the S-phase. 6C and D. I could not observe the BrdU-positive cells (around 35 %) on the flow cytometry panel for the dox-negative cells. Please clarified this point. 6B. Please could you quantify the fluorescence and provide the graph? Please include scale bare.

Response: In the revised version, we simplified the data of the immunofluorescence experiment into image B1 and counted cells in graph B2. The measurement of fluorescence intensity is not the intended parameter in this experiment. The intended parameter is the number of cells successfully incorporating BrdU showing replicative cells, which has increased in HN1 overexpressing (HN1-Venus expressing, Dox+) cells as compared to control (Dox-) cells. In the previous version of the paper, the mean number of BrdU+ cells was shown (Dox-=36/129 nuclei, Dox+=57/143 nuclei). The reviewer is right about the total number of BrdU+ cell percentages, which have been corrected in the revised manuscript. The reviewer miswrote figure 8B as 6B. For flow cytometric analysis, PC3-pCW57-HM-HN1 cells were used (because Venus interferes with the FITC channel). The dot blot shows the cells incorporating BrdU (labeled with anti-BrDU antibody) are higher in percentage. The raw data of the dot blot has been added to the supplementary information. The image in the C panel represents the overlay of cells with Dox+ cells exhibiting a higher percentage of BrdU+ cells. The graph showing the mean number of BrdU+ cell percentages has been added. Nevertheless, in both cases of different procedures and also different overexpression Cassettes (HN1-Venus and HM-HN1), the cells incorporating BrdU are enriched in HN1 stably overexpressing cells as compared to controls. The numerical data has been added to the figure legend. The scale bar has been added.

Figure 9: HN1 overexpression leads to cytoplasmic retention of Cdt1. 9A. “Cdt1 levels increased in the cytoplasm in HN1 overexpressed samples with a decrease in nuclear and chromatin fractions…..” please provide quantification of 3 independent experiments. I could not see this decrease on the image provided. Please authors should increase the quality of this image. 9B. Controls are missing for the interpretation of the experiment.

Response: We appreciate the observation and suggestion by the reviewer. The quantitative data has been added to the revised manuscript and has improved the quality of our manuscript. We have revised our claim that Cdt1 levels in the nucleus decrease with HN1 overexpression as quantitated data of experiments showed that Cdt1 levels only in chromatin fraction decreased with no change in the nuclear fraction. Meanwhile, the increased retention of Cdt1 in the cytoplasm and reduced loading onto chromatin upon HN1 overexpression do occur and bear no consequence on the findings of our investigations. The suggestion for quantification of blots has increased the quality and comprehension of our results.

The Acetyl-H4 blot is shown as the control of the chromatin samples. In PC3 cells, HN1 itself was not observed in the chromatin fraction of the lysates whereas in DU-145 ectopic expression of HN1 also did not alter, and therefore has not been shown. This information has been added to the text. Nevertheless, the main question was to determine the mechanism through which early S-accumulation may occur in HN1-overexpressing cells, and it has been shown in the form of reduced loading of Cdt1 onto chromatin and increased retention in the cytoplasm. Furthermore, the quantitated Cdt1 levels in chromatin fractions have been added to emphasize the findings.

General comment:
The manuscript should be proofread for English and editing … example line 272…

Response: The English language-related mistakes have been extensively proofread in the revised manuscript. The revised version is now improved for both grammatical and sentence-structure-related issues. We appreciate the assessment.

Reviewer 2 Report

The authors reported that HN1 strongly affected the cell cycle in prostate cancer cells and is considered to possess dual roles including S-phase progression and mitotic exit. The paper is well written and well discussed.

Author Response

The authors reported that HN1 strongly affected the cell cycle in prostate cancer cells and is considered to possess dual roles including S-phase progression and mitotic exit. The paper is well written and well discussed.

Response: We gladly appreciate the positive review and acceptance of our manuscript by the reviewer.

Reviewer 3 Report

At present manuscript submitted by Javed et al., is concentrated that HN1 is enriched in S-phase, phosphorylated in mitosis, and contributing to cyclin B1 degradation in prostate cancer cells. It is presented the novel reaction of HN1 in prostate cancer cells which is the potential beneficial target of prostate cancer cells. The following questions should be conceived.

1. Abstract: It should be widely revised with the present aim and results. The present abstract is unclear, such as why did the highlight on HN1/CyclinB1/Cdk1, etc.

2. Introduction: the present scientific hypothesis should be revised in logic.

3. The authors should be present the transfection efficiency in the manuscript. About the experimental designing: the authors carried on many inhibitor experiments. Interesting, the agonist should be supplemented that maybe get the novel results.

Author Response

At present manuscript submitted by Javed et al., is concentrated that HN1 is enriched in S-phase, phosphorylated in mitosis, and contributing to cyclin B1 degradation in prostate cancer cells. It is presented the novel reaction of HN1 in prostate cancer cells which is the potential beneficial target of prostate cancer cells.

Response: We appreciate the analysis of the manuscript and the positive response by the reviewer.

The following questions should be conceived.

  1. Abstract: It should be widely revised with the present aim and results. The present abstract is unclear, such as why did the highlight on HN1/CyclinB1/Cdk1, etc.

Response: The abstract has been revised according to the suggestion by the reviewer with more emphasis on the HN1/Cdh1/CyclinB1 axis. It has improved the quality and easy understanding of the manuscript.

  1. Introduction: the present scientific hypothesis should be revised in logic.

Response: The introduction of the manuscript has been expanded and revised to emphasize the logic and background relevant to the main hypotheses of the manuscript. Therefore, the overall readability and quality have been improved as a result of the revision.

  1. The authors should be present the transfection efficiency in the manuscript. About the experimental designing: the authors carried on many inhibitor experiments. Interesting, the agonist should be supplemented that maybe get the novel results.

Response: The plasmid transfection efficiency can be observed from very dense bands of HN1 (ectopic). HN1 silenced cells exhibit faint HN1 bands as compared to controls demonstrating the efficiency of siRNA. Moreover, the same siRNA was used in a previous study where transfection efficiency using FUGENE-HD was up to 70% using RT-PCR and western blotting in Prostate cancer cells. The same experimental protocols were applied here and immunoblotting validate the HN1 depletion. The densitometry analysis of transfection efficiency has been added to the manuscript and HN1 levels were silenced by up to 50% in the siRNA experiment. It has improved the quality of the work and we are grateful to the reviewer for the suggestion.

The rationale behind using several inhibitors was to show the possible upstream kinases of HN1. Our data showed that HN1 is phosphorylated only in mitosis and not in other phases of the cell cycle, therefore inhibitors to only kinases which could act in mitosis were used. Moreover, since the HN1 phospho-band disappeared upon both lambda-phosphatase and only GSK3-b and CDK1 inhibitors, therefore, it is logical to assume that these kinases are upstream regulators of HN1. The phospho-HN1 regulation and its role in mitosis can be studied in further studies. Our study is the first to characterize any potential post-translational modification of HN1 protein in any context. We are thankful to the reviewer for the comment.

Round 2

Reviewer 1 Report

In the revised manuscript, the authors tried to address the concerns.

However, some concerns were not completely addressed.